# TVCACHE: A Stateful Tool-Value Cache for Post-Training LLM Agents

**Abhishek Vijaya Kumar** [*1]  **Bhaskar Kataria** [*1]  **Byungsoo Oh** [1]  **Emaad Manzoor** [1]  **Rachee Singh** [1]

## Abstract

In RL post-training of LLM agents, calls to external tools take several seconds or even minutes, leaving allocated GPUs idle and inflating post-training time and cost. While many tool invocations repeat across parallel rollouts and could in principle be cached, naively caching their outputs for reuse is incorrect since tool outputs depend on the environment state induced by prior agent interactions. We present TVCACHE, a stateful tool-value cache for LLM agent post-training. TVCACHE maintains a tree of observed tool-call sequences and performs longest-prefix matching for cache lookups: a hit occurs only when the agent's full tool history matches a previously executed sequence, guaranteeing identical environment state. On three diverse workloads—terminal-based tasks, SQL generation, and video understanding—TVCACHE achieves cache hit rates of up to 70% and reduces median tool call execution time by up to $6.9\times$, with no degradation in post-training reward accumulation.

## 1. Introduction

Large language model (LLM) agents solve complex real-world tasks by interleaving text-based planning and reasoning with calls to *tools*: external functions that enable language models to perceive and mutate their environment (*e.g.,* by executing shell commands, querying databases, or invoking APIs) (Yang et al., 2024b; Team, Kimi et al., 2025; Zeng et al., 2025a; Agarwal et al., 2025).

The tool-calling capabilities of state-of-the-art agentic models are enabled by reinforcement learning (RL)-based post-training (Zhang et al., 2025; Zeng et al., 2025b; Jiang et al., 2025; Li et al., 2025; Feng et al., 2025). In each post-training iteration, the model generates several *rollouts*: token se-

---
[*]Equal contribution  [1]Cornell University. Correspondence to: Abhishek Vijaya Kumar <abhishek@cs.cornell.edu>, Bhaskar Kataria <bk478@cornell.edu>.

*Proceedings of the $43^{rd}$ International Conference on Machine Learning*, Seoul, South Korea. PMLR 306, 2026. Copyright 2026 by the author(s).

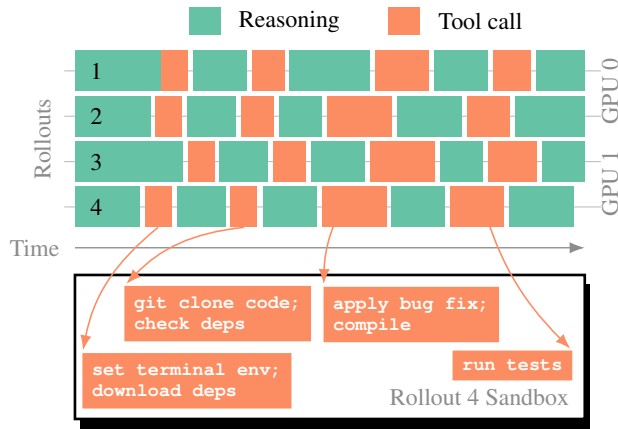

*Figure 1.* Illustration of 4 rollouts from an RL post-training iteration over time. The tokens generated in each rollout interleave between reasoning (green) and tool-calling (orange). Tools are executed in a sandbox environment and may mutate the sandbox state. In practice, tool execution comprises a significant portion of post-training time. TVCACHE maps tool outputs to tool calls in addition to a graph-based representation of the sandbox state to eliminate redundant tool executions and speed up post-training.

quences comprising both reasoning and tool calls, which are subsequently executed in isolated sandbox environments. Figure 1 illustrates 4 rollouts in 1 iteration of post-training a code-debugging agent, where the agent interacts with a terminal via tools to debug a codebase.

While modern accelerators can generate a token in milliseconds, tool calls generally take orders of magnitude longer to execute (*e.g.,* tools may compile entire codebases, run test suites, transcode video, or run web searches). Until a tool finishes executing in a rollout, the GPU resources allocated to that rollout remain idle. Our measurements across three representative benchmarks reveal that waiting for tool execution comprises 7–43% of the total rollout time on average, with tail cases exceeding 90%. This inefficiency translates into significantly higher post-training times and costs.

Our key observation is that tool calls are redundant across the rollouts of an RL-based post-training iteration. Since the rollouts in an iteration are induced by the same prompt, they frequently invoke similar sequences of tool calls (*e.g.,* cloning the same repository, building the same codebase, and running the same tests). We leverage this insight to cache tool call results and reuse them to reduce the latency of executing external tools in rollouts.

However, the *statefulness* of tool call executions makes caching challenging. Existing prompt or context caching methods (Bang, 2023; Zhu et al., 2024; Li et al., 2024; Couturier et al., 2025; Yu et al., 2025; Schroeder et al., 2025) find semantically similar contexts and reuse responses across them. These systems assume a stateless mapping from contexts to responses. Hence, they are inapplicable to post-training, wherein for each prompt, tool calls modify state.

Consider a code-debugging agent that, during a rollout, inspects a file by calling `cat foo.py`, applies a patch with another tool call, and then calls `cat foo.py` again. A stateless cache keyed only on tool names and arguments would return a stale cached value for the second `cat foo.py` call and thus silently degrade post-training.

Even semantically similar or identical contexts may correspond to different sandbox states when their sequences of tool calls differ. Reusing cached tool call results across such states can return incorrect outputs and degrade post-training. This challenge motivates our central research question:

**Q. How can we design a cache that correctly reuses work across stateful tool calls in RL post-training?**

To address this question, we propose TVCACHE: a stateful tool-value cache for post-training LLM agents. Our key insight is that caching stateful tool calls correctly requires reasoning not just about individual tool calls in isolation, but about the sequence or trajectory of tool calls that produced a given sandbox state.

Specifically, TVCACHE maintains a *tool call graph* (TCG)—a graph whose paths correspond to sequences of tool calls observed across rollouts in a training task. Each node in the TCG stores a tool call, its result, and (optionally) a snapshot of the sandbox state at the time of that call. When an agent invokes a tool during a rollout, TVCACHE performs a longest-prefix match against the TCG. If the agent's tool call trajectory so far exactly matches a cached path, TVCACHE returns the cached result directly. If the match is only partial, TVCACHE forks the sandbox cached at the final node of the longest matching prefix and executes only the remaining unmatched tool calls. This approach maximizes reuse of cached sandbox state while guaranteeing correctness.

Naively snapshotting every sandbox state would be prohibitively expensive. TVCACHE therefore employs a *selective snapshotting* policy that stores snapshots only when the expected cost of re-executing a tool exceeds the overhead of snapshot storage and retrieval. This policy naturally prioritizes caching expensive tool calls (*e.g.,* running full test suites and code compilations) over cheaper tool calls like file reads. Combined with proactive sandbox forking and efficiently coordinating concurrent access to cached snapshots for simultaneous rollouts, TVCACHE reduces the cache lookup overhead latency to milliseconds, while also enabling cache reuse across rollouts within an iteration, and even across post-training iterations.

We evaluate TVCACHE on three agentic post-training workloads, derived from terminal-bench (The Terminal-Bench Team, 2025), SkyRL-SQL (Liu et al., 2025), and the EgoSchema video question-answering dataset (Fan et al., 2025). Across these workloads, TVCACHE achieves cache hit rates up to 70% and reduces the median tool-call execution time by $6.9\times$ over post-training without caching, with no degradation in post-training reward accumulation, in self-hosted, cloud-hosted, and (Tinker) API-based post-training. We release TVCACHE as an open-source library at https://github.com/TVCache/TVCache.

## 2. Background and Motivation

### 2.1. The post-training agent loop

In each iteration of reinforcement learning (RL)-based post-training (Zhang et al., 2025; Zeng et al., 2025b; Jiang et al., 2025; Li et al., 2025; Feng et al., 2025), the agent generates multiple parallel *rollouts* conditioned on an input prompt representing an agentic task, such as "build and test the latest Linux kernel on this ARM machine". Rollouts interleave reasoning tokens (via which the agent plans and "thinks") with tool call tokens (via which the agent interacts with an isolated sandbox environment to carry out its task).

Tool calls are specially-formatted token sequences that map to external functions. Tool calls are executed inside an isolated *sandbox* that encapsulates mutable state for the rollout. Sandboxes may take various forms, such as a database instance (*e.g.,* for SQL agents), a Docker container (*e.g.,* for terminal agents), or a media store (*e.g.,* for video agents). Tool calls are the interface via which the agent both perceives and mutates the current sandbox state.

A rollout concludes either when the agent generates a special stop token, or when the configured maximum number of rollout tokens (*i.e.,* maximum rollout length) is reached. Upon rollout completion, a reward function assesses the success of that rollout and assigns it a reward. At the end of an iteration (*i.e.,* when all rollouts have concluded), the rewards across the rollouts are aggregated and used to update the agent's parameters. The parameter updates depend on the specific reinforcement learning algorithm, such as PPO (Schulman et al., 2017) or GRPO (Shao et al., 2024).

### 2.2. Performance overhead of tool execution

We now quantify the performance overhead of tool execution on different post-training workloads, with different model sizes, performed on different post-training infrastructures (self-hosted, cloud-hosted, and cloud API-based).

| Dataset | Agent | # Tasks | Hardware | # Epochs | # Rollouts | Max. Rollout Length |
|---|---|---|---|---|---|---|
| terminal-bench (easy) | Qwen3-4B-Instruct-2507 | 51 | 2×A100 80G | 10 | 8 | 2048 |
| terminal-bench (med) | Qwen3-4B-Instruct-2507 | 95 | 2×A100 80G | 10 | 8 | 2048 |
| terminal-bench (easy) | Qwen3-14B-Instruct | 51 | 8×A100 80G (cloud) | 10 | 4 | 2048 |
| terminal-bench (med) | Qwen3-14B-Instruct | 95 | 8×A100 80G (cloud) | 10 | 4 | 2048 |
| SkyRL-SQL | Qwen2.5-Coder-7B-Instruct | 653 | 8×A100 80G (cloud) | 10 | 5 | 3000 |
| EgoSchema | Qwen3-30B-A3B-Instruct-2507 | 100 | Tinker API (cloud) | 5 | 8 | 32768 |

*Table 1.* Post-training workload datasets and configurations.

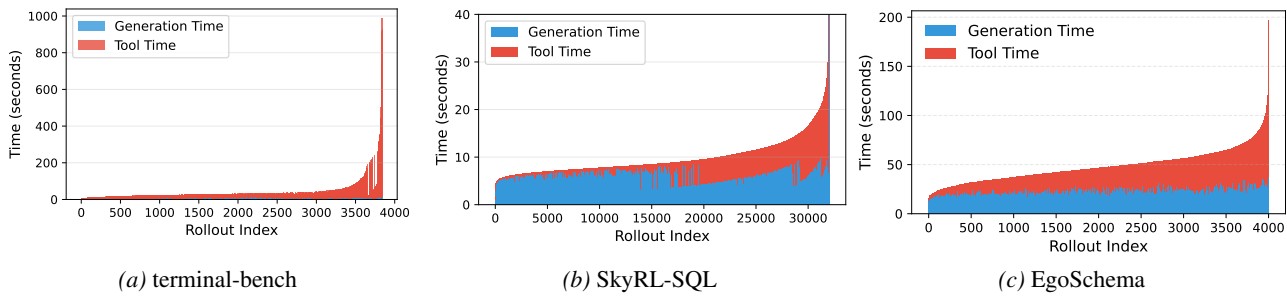

*(a)* terminal-bench      *(b)* SkyRL-SQL      *(c)* EgoSchema

*Figure 2.* Wall-clock time taken by each rollout (sorted by the total wall-clock time) for reasoning token generation and tool call execution in the (a) terminal-bench, (b) SkyRL-SQL, and (c) EgoSchema post-training workloads (see Table 1 for workload details).

Specifically, we post-train Qwen (Yang et al., 2024a) open-weight models on 3 agentic workloads: (1) terminal-bench (The Terminal-Bench Team, 2025), comprising terminal-based tasks executed in Docker containers; (2) SkyRL-SQL (Liu et al., 2025), comprising data processing tasks (via SQL) on a cloud-hosted database; and (3) EgoSchema (Fan et al., 2025), comprising 100 video processing and understanding tasks[1]. Our complete post-training workload configurations are listed in Table 1.

Our selected workloads comprise different types of tool calls. In the terminal-bench workload, tool calls are Linux terminal commands such as package installations, code compilations, and test runs. In the SkyRL-SQL workload, tool calls are SQL queries that read from a Google cloud-hosted database instance. In the EgoSchema workload, tool calls are remote procedure calls (RPCs) that perform video processing tasks like extracting frames and generating captions.

We instrument the post-training to measure the wall-clock time of reasoning token generation and tool call execution within each rollout. Figure 2 shows the time spent in token generation versus tool execution per rollout across the three benchmarks (rollouts sorted by their total wall-clock time).

Across the 3 workloads, between 7% and 43% of the rollout time is consumed by tool execution; until the tool completes execution, the GPU resources processing that rollout remain idle. For terminal-bench (Figure 2a), tool calls consume 43% of the rollout time on average, and the 99th percentile tool execution time consumes over 92% of the rollout time. For SkyRL-SQL, tool calls consume 7% of the roll-

out time on average, and the 95th percentile tool execution time consumes 43% of the rollout time. For EgoSchema, tool calls consume 32% of the rollout time on average. In summary, our results across workloads, models, and hardware/software infrastructures show that tool execution is a key source of inefficiency in post-training.

### 2.3. Opportunities, Challenges, and Our Contributions

**Opportunity to cache tool-call results.** Our key insight is that, since RL-based post-training produces several rollouts in parallel for each input prompt (or task) in an iteration, many tool calls are redundant across rollouts. For example, rollouts for a task often build the same codebase, run the same test suite, or query the same database table. By identifying when an agent repeats a previously-executed sequence of tool calls, we can reuse prior results to amortize tool execution time and improve post-training efficiency.

**Challenge I: Tool Statefulness.** However, tools can be stateful *i.e.*, the output of a tool is not determined solely by its arguments, and also depends on the entire history of prior mutations of the sandbox in which the tools execute (which determines the sandbox's state). This dependency on sandbox state makes caching tool calls fundamentally more challenging than caching prompts or stateless functions (Bang, 2023; Zhu et al., 2024; Yu et al., 2025; Schroeder et al., 2025). Therefore, a correct tool cache must only reuse the result of a tool call when the sandbox state exactly matches what it was when the result was first computed.

**Challenge II: Caching Efficiently and Concurrently.** On a cache miss, an efficient cache should construct the sandbox state required to execute the current tool call by reusing past

---

[1]We randomly sample 100 tasks from the 500 in the dataset due to computational budget constraints.

sandbox states. In principle, we could snapshot the entire sandbox after every tool call and store it as part of the cache value. However, naively snapshotting the sandbox state after *every* tool call (and restoring snapshots frequently) adds significant overhead. Finally, rollouts executing in parallel benefit from concurrent access to cached sandboxes, but enabling cache concurrency efficiently is non-trivial.

**Contributions.** We propose TVCACHE to address these challenges and achieve the following goals:

- *State-aware tool caching.* TVCACHE reuses a cached tool result only when the rollout has reached the same sandbox state as the cached key.
- *Low-overhead cache lookups.* TVCACHE organizes the cache as a tool call graph, where each root-to-node path represents a sequence of tool calls. Thus, cache lookups reduce to efficient longest-prefix matches.
- *Selective sandbox snapshotting.* TVCACHE stores sandbox snapshots selectively only when the expected cost of reconstructing a sandbox from a sequence of tool calls exceeds the snapshotting overhead.
- *Concurrent cache-sharing across parallel rollouts.* TVCACHE supports highly parallel post-training by (i) sharding the cache, and (ii) coordinating safe concurrent access to shared prefixes in the sharded cache.

## 3. TVCACHE Design

TVCACHE organizes the cache as a tool call graph (§3.1), which enables reconstructing sandbox states from tool call sequences and reduces cache lookups to efficient longest prefix matches (§3.2). TVCACHE further augments the TCG with serialized sandbox snapshots only when the expected cost of sandbox reconstruction exceeds that of snapshotting, a policy we refer to as selective snapshotting (§3.3). We now discuss the TVCACHE design in detail.

### 3.1. Tool Call Graph (TCG) Structure

For each task (or prompt) $p$, TVCACHE maintains a tool call graph $\mathcal{G}(p) = (\mathcal{V}(p), \mathcal{E}(p))$ that is shared across the parallel rollouts for that task and reused across post-training iterations. Each node $v \in \mathcal{V}(p)$ represents the tuple $(t, r, s)$ where $t$ is the name of the tool and its arguments (*i.e.,* the tool descriptor), $r$ is the tool execution result, and $s$ is the serialized sandbox snapshot immediately after tool execution (where $s$ is stored selectively and may be null). Each directed edge $(u, w) \in \mathcal{E}(p)$ represents a pair of subsequent tool calls and their associated results and snapshots.

$\mathcal{G}(p)$ is initialized with a dummy root. As each rollout for $p$ progresses, each of its tool calls either (a) misses the cache and creates a node in $\mathcal{G}(p)$ and a directed edge in $\mathcal{E}(p)$ from the previous tool call (or from the root if there is no previous

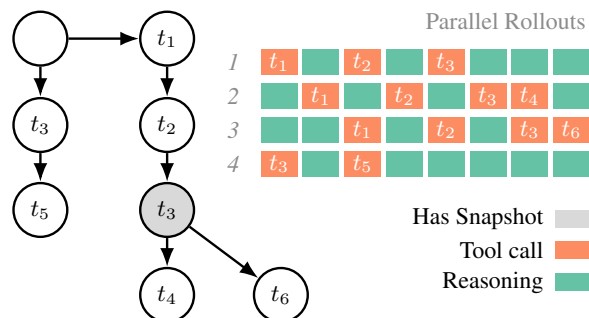

*Figure 3.* Tool Call Graph (TCG) $\mathcal{G}(p)$ constructed from 4 rollouts.

tool call), or (b) hits the cache and has no effect on $\mathcal{G}(p)$. Figure 3 illustrates a TCG constructed from four parallel rollouts invoking 6 unique tool descriptors in total. Figure 9 in Appendix A shows a real TCG from our experiments.

### 3.2. Cache Lookups, Hits, and Misses

On each tool call $t_j$ preceded by tool calls $t_1, t_2, \ldots, t_{j-1}$ in the rollout, TVCACHE performs a cache lookup by searching for the *longest prefix match (LPM)* of the sequence $q = \{t_1, t_2, \ldots, t_j\}$ in the TCG (a $\log|\mathcal{V}(p)|$ time operation, typically taking a few milliseconds).

A matched prefix $\tilde{q} = q$ corresponds to a *cache hit*, and TVCACHE returns the tool execution results from the matched TCG node $t_j$. Longest prefix matching guarantees that the returned result is identical to that from executing $t_j$ in a sandbox that has been mutated by the same sequence of tools as the current rollout, guaranteeing correctness. In Figure 3, the calls to $t_1$, $t_2$, and $t_3$ in rollouts 2 and 3 are cache hits, due to rollout 1 having called these tools previously.

A matched prefix $\tilde{q} \neq q$ is a *cache miss*. $\tilde{q}$ is the longest prefix of $q$ that has an exact match in the TCG and $q_{\text{unmatched}}$ is the remaining unmatched suffix of $q$. If the final node $t'$ of $\tilde{q}$ has a sandbox snapshot, TVCACHE executes the tools in $q_{\text{unmatched}}$ sequentially in that snapshot and returns the result of the final tool execution, in addition to appending it to the TCG. If the final node $t'$ of $\tilde{q}$ does not have a sandbox snapshot, TVCACHE executes the full tool sequence $q$ in a new sandbox and returns the result of the final tool execution.

Not all tools modify state. If stateless tools are annotated as such, TVCACHE can further improve cache reuse by skipping them in longest-prefix matching, effectively matching prefixes over only the stateful tool calls in the TCG. We analyze this optimization in Appendix B.

**Non-determinism in tool call results.** Non-determinism in tool call results can arise for two reasons. First, the unobservable state outside the sandbox may change over time, *e.g.,* updates to Google's search index will change the result of a web-search tool call. Second, the tool call itself may be stochastic, *e.g.,* a data-analysis tool that performs

random sampling may return different samples for the same input. To address non-determinism, TVCACHE adds a time to live (TTL) mechanism that invalidates TCG entries after a configurable duration. TVCACHE records time of insertion for each node in the TCG, and periodically prunes nodes whose age exceeds the configured TTL duration. If an expired node has outgoing edges, TVCACHE recursively prunes the entire subtree rooted at that node.

### 3.3. Selective Sandbox Snapshotting

A naive hashtable-based cache could include the full sandbox state in each cache key, enabling O(1) lookups compared to TVCACHE's O($\log N$) prefix matching. However, the memory overhead of storing a separate sandbox snapshot for every cached entry is prohibitive. Even with TVCACHE's TCG representation, naively snapshotting the sandbox after every tool call remains too expensive.

Instead of storing a snapshot at each node, TVCACHE compares the time spent executing that node's tool call with the overhead of snapshotting *i.e.,* the cost to serialize and later restore a sandbox snapshot. TVCACHE stores a snapshot only if executing the tool call is more expensive than the snapshotting overhead. In practice, this policy prioritizes snapshots for long-latency tool calls (*e.g.,* running a full test suite or compiling a large codebase) and avoids snapshotting for inexpensive actions (*e.g.,* reading a small file).

**Sandbox forking.** TVCACHE treats every sandbox in the TCG as immutable. When a rollout resumes from a cached sandbox, TVCACHE *forks* or copies the cached sandbox and continues to execute tools in it. Since forking in the critical path incurs snapshot restoration latency, TVCACHE employs three strategies to optimize sandbox forking:

- **Proactive forking.** TVCACHE proactively forks sandboxes to reduce future forking overheads. Before the start of a training step, TVCACHE creates multiple root sandboxes, *i.e.,* clean sandboxes that rollouts can use as an initial state. Root sandboxes are created and kept warm proactively so that, when a rollout begins, TVCACHE can supply a ready-to-use sandbox without paying sandbox start-up latency. In addition, for any TCG node that already has a saved snapshot, TVCACHE pre-creates a forked copy of the corresponding sandbox. These copies are kept ready so that, if a rollout later incurs a cache miss and must resume from the longest-prefix match node, execution can start immediately in the correct sandbox. This proactive warmup allows many cache misses to be handled without expensive synchronous container creation.
- **Reactive forking.** When a rollout incurs a cache miss but still has a longest-prefix match in the TCG, TVCACHE first checks if a background thread has produced a forked sandbox for the node. If such a fork exists, the rollout resumes with negligible delay. Otherwise, the rollout

forks the sandbox on the critical path.
- **Background instantiation.** When a tool call finishes and is deemed sufficiently expensive, TVCACHE snapshots the resulting sandbox state on the critical path of the rollout. However, it offloads the instantiation of the sandbox (*i.e.,* producing the forked sandbox for reuse) to a background thread. That thread instantiates the forked sandbox from the snapshot and later attaches it to the corresponding node in the TVCACHE graph. In effect, TVCACHE combines synchronous and asynchronous sandbox creation: sandboxes needed immediately are created on the critical path, while sandboxes that are primarily valuable for future reuse are instantiated in the background.

**Bounding number of cached sandboxes.** TVCACHE bounds the total number of sandboxes stored in the cache. Each task can specify a budget for active sandboxes. When the number of stored sandboxes exceeds this budget, TVCACHE prunes the least useful ones by evicting subtrees with low expected reuse. Eviction decisions take into account both the depth of the node and the number of children to avoid deleting subtrees that capture common prefixes.

### 3.4. TVCACHE Implementation

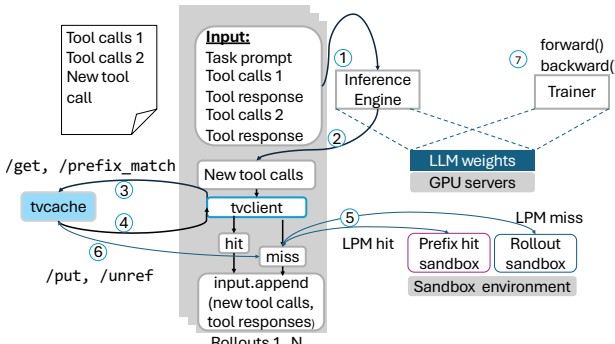

*Figure 4.* TVCACHE's architecture. The agent generating the rollout interacts with TVCACHE through `ToolCallExecutor` and `ToolCallEnvironment` in the `tvclient` library.

TVCACHE adopts a server-client architecture as shown in Figure 4. The TVCACHE server is an HTTP service that manages the TCG and associated sandbox snapshots. It exposes endpoints for all operations on the cache: inserting new sequences (`PUT /put`), retrieving exact matches (`GET /get`), performing longest-prefix matches (`POST /prefix_match`), storing and retrieving test results and visualizing the TCG. Each endpoint manipulates the underlying graph through a thread-safe API. The server persists TCG snapshots periodically to disk, and collects cache-hit statistics, required by the pruning policy.

The TVCACHE client library provides a `ToolCallExecutor` class that the RL training loop implements. Before executing a tool call, the rollout serializes the call (tool name and arguments) into a string,

concatenates it with previous tool calls, and passes the resulting sequence to the executor. The executor queries the cache (`/get`) for an exact match of this sequence. On a hit, it returns the cached value. Otherwise, the rollout executes the tool call in a forked sandbox returned by `prefix_match`.

**Concurrency Control.** TVCACHE relies on reference counting to avoid eviction of sandboxes while they are being forked. After detecting a longest prefix match (LPM), the TVCACHE server increments the reference count of the sandbox at the end of the prefix before returning. The client decrements the count after forking and the eviction policy evicts only sandboxes with zero references (Figure 4).

**Sandbox lifecycle.** TVCACHE abstracts sandbox management through the `ToolExecutionEnvironment` class. Each dataset implements this class by defining four methods: `start`, `stop`, `fork`, and `execute`. The executor uses objects of this class to interact with the sandbox.

**Integration with RL post-training frameworks.** This design makes it easy to integrate TVCACHE with various RL frameworks. We have integrated it with locally-hosted and cloud-hosted post-training using veRL and the RL-as-a-service offering Tinker for our evaluation.

## 4. Evaluating TVCACHE

In this section, we evaluate TVCACHE by post-training agents for three different domains: terminal tasks (terminal-bench), database queries (SkyRL-SQL), and video understanding (EgoSchema). We use the same workloads and hardware/software configurations listed in Table 1. We provide details about reward functions, batch sizes, training frameworks, and loss functions in Appendix C and all the prompts (tasks) in Appendix G.

### 4.1. terminal-bench workload

We post-train different models using GRPO (Shao et al., 2024) for the terminal-bench (easy) and terminal-bench (medium) tasks, with different post-training configurations and hardware as specified in Table 1. The terminal-bench workload provides a Dockerfile for each task. Hence, we use Docker containers as sandboxes in which the tool calls (bash commands) execute. For the terminal-bench (easy) tasks, we run the Docker containers on a self-hosted server with 128 cores and 128 GB of memory. For the terminal-bench (medium) tasks, we run the Docker containers on a cloud server with 30 CPUs and 200 GB of memory.

**Scaling sandbox creation and snapshotting.** For a batch size of $B$ tasks and with $R$ rollouts per task, TVCACHE creates $BR$ root containers at the start of post-training. During post-training, TVCACHE creates a background container

for each internal node of the TCG for proactive forking. Thus, a post-training run creates hundreds of containers in total, and we found terminal-bench's container management system (Harbor Framework Team, 2026) unable to scale. In Appendix E, we describe our modifications to this container management implementation that scales to hundreds of containers. We use Docker's native checkpointing functionality to snapshot and restore sandboxes.

**Results.** Figure 5b reports cache hit rates by epoch. On easy tasks, TVCACHE achieves an average hit rate over epochs of 20.2% for the Qwen3-4B-Instruct-2507 agent and 25.3% for the Qwen3-14B-Instruct agent. On medium tasks, the hit rates are 14.2% and 16.7%, respectively. Hit rates increase with epochs as the TCG grows and has more branches to find cache hits. Larger models achieve higher hit rates because they tend to repeat tool calls.

| Model Size | Task Difficulty | No Cache (s / call) | Cache (s / call) | Speedup |
|---|---|---|---|---|
| Qwen3-4B-Instruct | Easy | 8.67 | 1.40 | 6.18× |
| Qwen3-4B-Instruct | Med | 18.68 | 2.70 | 6.92× |
| Qwen3-14B-Instruct | Easy | 8.07 | 2.35 | 3.44× |
| Qwen3-14B-Instruct | Med | 36.23 | 6.53 | 5.55× |

*Table 2.* Median *per-tool-call* execution time and speedup

We now evaluate whether cache hits translate to speedups in practice. In Table 2, we report the median tool call execution time with and without TVCACHE, showing a reduction in median tool call execution time by up to 6.9×. In Appendix F, we provide the complete tool execution and rollout time distribution across all the configurations from Table 2.

### 4.2. SkyRL-SQL workload

We post-train a Qwen2.5-Coder-7B-Instruct agent for the SkyRL-SQL workload using GRPO, configured as specified in Table 1. We use a Google cloud virtual machine with 8 CPUs and 32 GB of memory running SQLite as the sandbox, with a median round-trip network latency of 55.8 milliseconds from the post-training server. The SkyRL-SQL workload comprises stateless tool calls (SQL read queries); hence, sandbox snapshotting is a no-op for this workload.

**Results.** Figure 5a shows cache hit rates by epoch. TV-CACHE achieves an average hit rate of 33.11%.

Each cache hit reduces the tool execution time from roughly 56.6 milliseconds to 6.5 milliseconds, a speedup of 8.7×. Given the average cache hit rate of 33.11% over epochs, this translates to an expected tool call speedup of 2.9×.

### 4.3. EgoSchema workload

For the EgoSchema video understanding workload, we post-train a Qwen3-30B-A3B-Instruct-2507 agent using the Tin-

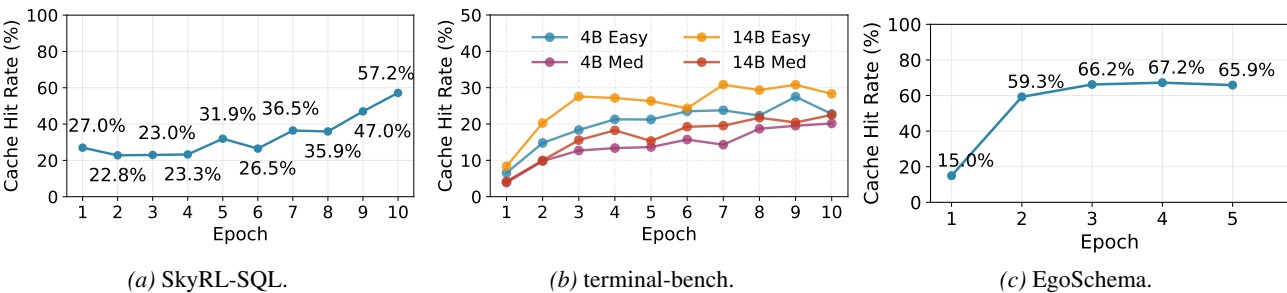

*(a)* SkyRL-SQL.    *(b)* terminal-bench.    *(c)* EgoSchema.

*Figure 5.* Cache hit rates over post-training epochs for three workloads. TVCACHE exhibits high hit rates which increase over post-training epochs due to the tool call graph growing and branching further. Hit rates in the terminal-bench workload range from 15% to 32%. Hit rates in the SkyRL-SQL workload range from 27.0% to 57.2%. Hit rates in the EgoSchema workload range from 34% to 73.9%.

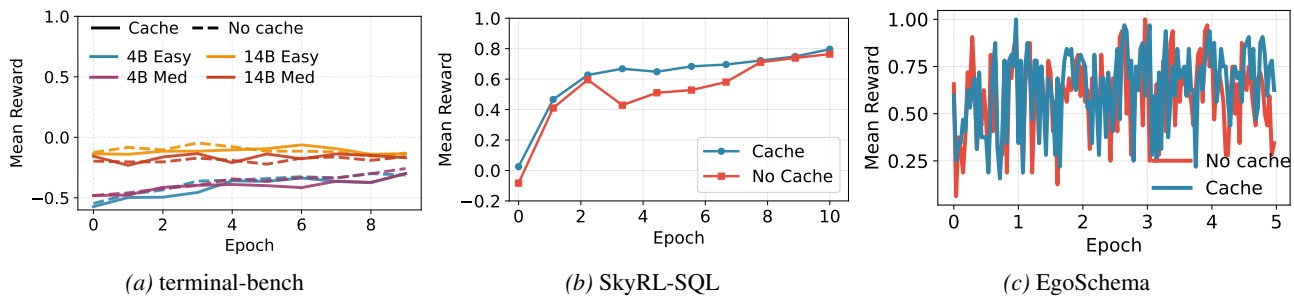

*(a)* terminal-bench    *(b)* SkyRL-SQL    *(c)* EgoSchema

*Figure 6.* TVCACHE produces reward curves that closely match the reward curves without caching for all workloads.

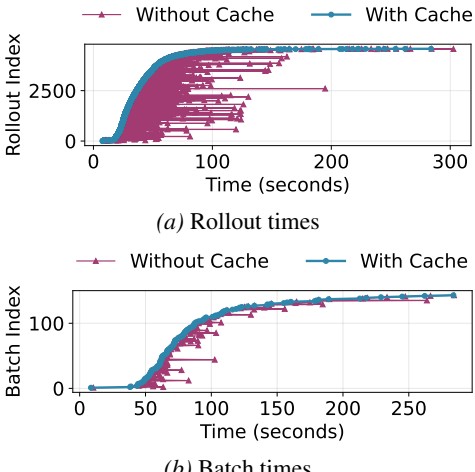

*(a)* Rollout times

*(b)* Batch times

*Figure 7.* Total time of each (a) rollout, and (b) batch during post-training with (blue) and without (purple) TVCACHE. Rollouts and batches are sorted by their total execution times with TVCACHE. TVCACHE consistently reduces rollout and batch execution times.

ker API configured as specified in Table 1. We post-train using policy gradients (Williams, 1992; Sutton et al., 1999) with importance sampling. Though the agent is post-trained in the Tinker cloud, we execute the tools called in rollouts on a self-hosted server with an NVIDIA L40S GPU with 48 GB of video memory, 128 CPU cores, and 128 GB RAM. We use a directory on the server for each task as the sandbox state. To fork a sandbox state, we make a copy of the task's

directory. The agent has access to tools that we adapted from VideoAgent (Fan et al., 2025) for solving the task. Appendix D describes the tools in more detail.

**Results.** Figure 5c reports the cache hit rate by epoch, and shows that TVCACHE achieves an average hit rate of 55% over epochs. Figure 7 shows how cache hits translate into reductions in the rollout and batch execution time, demonstrating a consistent reduction in execution time across rollouts and batches. Because batch time is determined by the slowest rollout, the overall batch completion time savings are lower (Figure 7b vs. 7a). For tool calls that invoke the OpenAI API, TVCACHE reuses past tool call results and reduces token usage by $1.67\times$ in total and up-to $3\times$ for certain tasks. We provide a detailed breakdown of improvements across tools and how TVCACHE skips stateless tools during LPM in Appendix D.

### 4.4. Testing for post-training performance degradation

We now evaluate whether TVCACHE degrades post-training performance in terms of the reward accumulated by the agent over epochs. Since TVCACHE is an exact cache (with guaranteed correctness), we do not anticipate any performance degradation. Figures 6a, 6c and 6b validate this empirically and show that reward curves remain similar with and without TVCACHE across model sizes (4B, 14B and 30B) for all workloads.

## 4.5. Comparison with a stateless cache

A natural baseline for TVCACHE is a stateless cache that uses only the tool name and arguments as the key in the cache, without including the sandbox state from which the tool was invoked. A stateless cache can increase the cache hit rate, but produces incorrect cache responses for stateful environments because identical tool calls can produce different outputs after the agent modifies the sandbox.

We evaluate this baseline on the terminal-bench dataset (easy tasks) by replaying tool calls in a sandbox with the correct state to verify whether stateless cache hits return correct outputs. As shown in Table 3, more than 25% of stateless cache hits are incorrect, returning stale tool call outputs.

| Model | Incorrect Stateless Hits |
|---|---|
| Qwen3-4B-Instruct | 27.4% |
| Qwen3-14B-Instruct | 26.1% |

*Table 3.* Fraction of stateless cache hits on terminal-bench (easy tasks) that return incorrect tool call output.

## 4.6. Sensitivity to rollout parallelism

TVCACHE benefits from repeated tool calls both within an epoch and across epochs. To evaluate whether the cache is still useful when fewer rollouts are generated per task, we rerun terminal-bench and SkyRL-SQL while varying the number of parallel rollouts per task. Table 4 shows that TVCACHE continues to obtain cache hits even with one rollout per task, because the TCG persists across post-training iterations and captures reuse across epochs.

| Rollouts / Task | terminal-bench | SkyRL-SQL |
|---|---|---|
| 1 | 10.1% | 23.4% |
| 2 | 13.1% | 23.3% |
| 4 | 17.0% | 27.8% |
| 8 | 20.2% | 33.4% |

*Table 4.* Average cache hit rate as the number of parallel rollouts per task varies. The 8-rollout setting matches the GRPO configuration used in the main experiments.

As expected, hit rates generally increase with more parallel rollouts because there are more opportunities for reuse within each batch. However, the one-rollout setting still achieves hit rates of 10.1% on terminal-bench and 23.4% on SkyRL-SQL, showing that TVCACHE is not specific to highly parallel GRPO configurations.

## 4.7. Microbenchmarking TVCACHE

**Cache hit latency.** We ran experiments on a 128-core machine with 128 GB RAM, using asynchronous clients on the same host to populate the cache with 8,000 distinct keys and generate load at controlled rates, eliminating network effects and isolating server-side latency. A single TVCACHE server achieves a P95 cache get latency of 3.3 ms at 256 RPS, but saturates at 512 RPS, where P95 latency exceeds 1 s (Figure 8a). Since each task's TCG is independent, TVCACHE shards the cache servers by task ID, enabling near-linear throughput scaling. As shown in Figure 8a, sharding preserves low tail latency under high load: with 16 shards, TVCACHE sustains 4096 RPS and keeps P95 latency at 6.1ms.

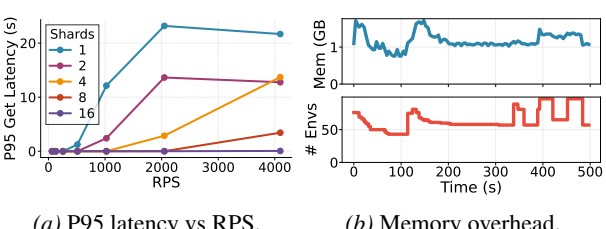

*(a)* P95 latency vs RPS.     *(b)* Memory overhead.

*Figure 8.* TVCACHE scaling characteristics.

**Cache-miss overhead.** For each request, TVCACHE first performs a cache lookup and falls back to baseline tool execution. Since the P95 cache get latency is under 10 ms, cache misses incur less than 10 ms of additional overhead.

## 4.8. Proactive forking memory overhead

Figure 8b shows the memory footprint of TVCACHE when training Qwen3-4B-Instruct on easy terminal-bench tasks for five steps. Memory usage remains low at around 1 GB, peaking at 2 GB. With a batch size of 4 and 8 rollouts per prompt, TVCACHE launches 32 containers for the next step while the current step is still running. Memory rises briefly during container creation at the start of each step and drops as rollouts finish and sandboxes are cleaned up, producing the observed spikes. Overall the memory use increases because TVCACHE caches 36 sandboxes in the TCG.

## 5. Related Work

Our work is related to three lines of research: post-training language models for tool-use, semantic caching for language model inference, and prefix caching in LLM serving systems. We detail related work in each of these lines below.

### 5.1. Post-training language models for tool-use

Tool-use or function-calling capabilities turn language models from text generators into agents that can interact with external environments. Early work trained language models to use tools via supervised fine-tuning on curated tool-use traces (Schick et al., 2023). Recent approaches train language models to use tools via reinforcement learning, which improves their ability to generalize to a wide variety of previously-unseen tools (Zhang et al., 2025; Zeng et al.,

2025b; Jiang et al., 2025; Li et al., 2025; Feng et al., 2025).

Tool calls incur substantial time and monetary costs, and emerging work has begun to address the effects of this on post-training efficiency. Kimi K2 amortizes tool call costs by simulating tool responses with another language model (Team, Kimi et al., 2025). OTC-PO (Wang et al., 2025) incorporates the cost of tool calls into the reinforcement learning reward function. VERLTOOL (Jiang et al., 2025) reduces tool calling time via asynchronous and parallel tool call execution. Our work is complementary; we propose a state-aware tool cache for post-training that reuses the results of expensive tool calls.

### 5.2. Semantic caching for language model inference

Semantic caching (also called context caching and prompt caching) methods accelerate language model inference by retrieving cached responses to identical or semantically similar prompts instead of generating them (Bang, 2023; Hu et al., 2024; Gim et al., 2024; Zhu et al., 2024; Li et al., 2024; Gill et al., 2025; Yang et al., 2025a;b; Couturier et al., 2025; Yu et al., 2025; Gu et al., 2025; Schroeder et al., 2025). Various commercial and open-source LLM inference systems now employ semantic caching to reduce latency and costs (Zheng et al., 2024; Redis).

Semantic caching assumes a stateless prompt–response mapping and thus does not directly apply to caching tool calls that may modify state during post-training. However, the approaches to identify similar prompt prefixes proposed by semantic caching methods can also be applied to identify similar tool-call trajectories and to design "fuzzy" tool-value caches; we delegate exploring this direction to future work.

### 5.3. Prefix caching in LLM inference

LLM serving systems, including open-source inference engines and hosted APIs, increasingly support prefix or prompt caching to reduce the computation of repeated prompt prefixes (vLLM Project, 2026; Zheng et al., 2024; SGLang Team, 2026; Anthropic; OpenAI). Prefix caching in inference systems reuses the key-value cache for common prefix tokens across prompts that share the same prefix. Agents can benefit from this optimization while generating tokens across rollouts. However, prefix caching only improves token generation efficiency and the agents still need to execute the tool calls generated during rollouts, even when those tool calls are identical across rollouts. TVCACHE is complementary to prefix caching and addresses this remaining inefficiency during rollout generation by accurately reusing tool call results across rollouts with a stateful cache.

## 6. Conclusion

We presented TVCACHE, a stateful tool-value cache that addresses a key inefficiency in RL-based post-training of LLM agents: the time spent waiting for external tool execution while GPU resources remain idle. By organizing cached tool calls into a tool call graph and using longest-prefix matches for cache lookups, TVCACHE correctly handles the statefulness of tool executions. TVCACHE returns cached results only when the agent's tool history guarantees an identical sandbox state. Using selective sandbox snapshotting and proactive forking, TVCACHE scales to large batch sizes. TVCACHE achieves cache hit rates up to 70% and reduces median tool-call execution time by up to $6.9\times$ across terminal-based, SQL, and video understanding workloads, with no degradation in post-training rewards. TVCACHE represents a step toward efficient infrastructure for training LLM agents capable of using a larger number of tools to accomplish more complex tasks.

## Impact Statement

This paper presents work whose goal is to advance infrastructure support for Machine Learning. We build a caching system to reduce computational waste during reinforcement learning (RL) post-training of LLM agents. By eliminating redundant external tool calls that can take seconds to minutes, our system tackles an efficiency bottleneck in agent training pipelines. Reducing redundant computation directly decreases energy consumption and carbon emissions associated with RL post-training. By reducing the cost of agent post-training, our system may accelerate the development of capable agents, including those with potential for misuse. However, as a systems optimization for existing training systems, our work does not introduce new capabilities or change the fundamental safety properties of the agents.

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

## A. Tool Call Graph (TCG)

Figure 9 shows an example of a Tool Call Graph from one of the terminal-bench training runs.

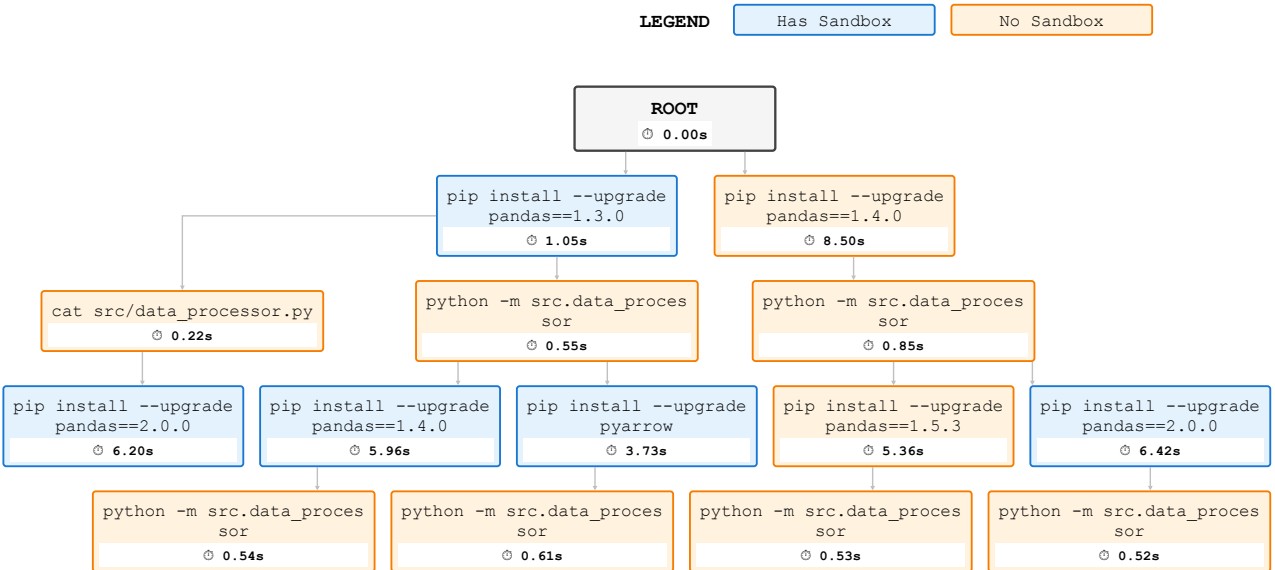

*Figure 9.* Example Tool Call Graph (TCG) $\mathcal{G}(p)$ from terminal-bench training. Each node represents a tool call with its result and optionally a sandbox snapshot. Edges represent sequential tool calls from a rollout. The graph structure enables efficient longest prefix matching for cache lookups and demonstrates how TVCACHE reuses tool execution results across parallel rollouts during RL training.

## B. Stateful Prefix Matching

TVCACHE defines an abstract method `will_mutate_state()` in the `ToolExecutionEnvironment` interface that allows developers to annotate whether a tool invocation may modify the sandbox state. By default, TVCACHE conservatively assumes that every tool mutates state, which is safe to assume when the tool space is large and a tool's behavior depends on the current state of the sandbox, *e.g.,* bash programs. Sandboxes like the EgoSchema workload have a limited set of tools and only two of these tools (preprocess, load_video) modify the sandbox state. In such cases, the cache hit rate and LPM hit rate increase significantly if we skip tool calls that execute without modifying the sandbox state.

In this section, we formally show that skipping non-state-mutating tools in the prefix during LPM preserves correctness.

**Assumption 1 (State determines tool output).** The output of any tool call $T$ depends only on the current sandbox state and the arguments of $T$.

**Assumption 2 (Correct state annotation).** For any tool $S$ such that `will_mutate_state()` returns false, executing $S$ does not modify the sandbox state.

**Correctness of stateful prefix matching.** Let a tool call prefix of a rollout consist of an interleaving of tools modifying the state $F_1, F_2, \ldots, F_N$ and tools preserving the state $S_1, S_2, \ldots S_N$. Let $\mathcal{P}$ denote the full prefix and let $\mathcal{P}' = \langle F_1, F_2, \ldots, F_N \rangle$ be the subsequence obtained by removing all $S$ tools while preserving the order of the $F$ tools. Then, $\mathcal{P}$ and $\mathcal{P}'$ correspond to the same sandbox state in the TCG. Consequently, longest-prefix matching (LPM) performed over only the $F$ tools is correct.

Assume that executing $\mathcal{P}$ and $\mathcal{P}'$ from the same initial sandbox state results in different sandbox states. Since $\mathcal{P}'$ is obtained from $\mathcal{P}$ by removing only state-preserving tools, the two executions differ only in the execution of tools $S$.

Let $S_j$ be the first such tool whose execution causes the sandbox states to diverge. This implies that executing $S_j$ modified the sandbox state. However, by Assumption 2, any tool $S_j$ marked as state-preserving does not mutate the sandbox state. This contradiction implies that no such divergence can occur.

Therefore, executing $\mathcal{P}$ and $\mathcal{P}'$ must result in identical sandbox states. By Assumption 1, the output of any subsequent tool call depends only on the current sandbox state. Hence, for the purposes of TCG construction and LPM, the presence or

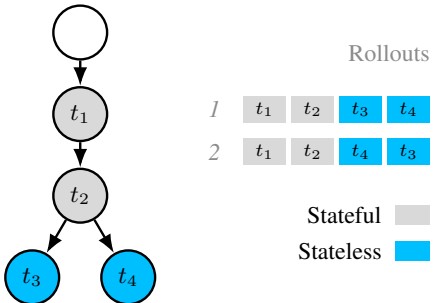

*Figure 10.* TCG with stateful prefix matching. State-preserving tools $t_3$ and $t_4$ are indexed as children of $t_2$ even though they execute in different orders in different rollouts.

absence of tools that preserve the state in the prefix does not affect either sandbox state or the suffix tool outputs.

As a result, TVCACHE can safely perform longest-prefix matching in the TCG using only the subsequence of state-modifying tools, while preserving correctness of cache hits and sandbox reuse.

**Cache hits.** When searching for a cache hit, the system removes all stateless tools from the prefix before comparing rollouts. In Figure 10, both rollouts share the same stateful prefix $(t_1, t_2)$ shown in grey, but differ in their stateless tool calls (blue). Rollout 1 executes $t_3$ then $t_4$ and after some time rollout 2 executes $t_4$ then $t_3$. When performing prefix matching for $\langle t_1, t_2, t_3, t_4 \rangle$ and $\langle t_1, t_2, t_4 \rangle$ the system filters both prefixes to their stateful subsequence $\langle t_1, t_2 \rangle$, recognizing the prefixes as equivalent since the stateless tools $t_3$ and $t_4$ do not modify the sandbox state. Without this optimization, rollouts 1 and 2 would be treated as having diverged after $t_2$, missing the reuse opportunity despite reaching identical sandbox states.

**Implication for Prefix Matching.** Figure 10 illustrates why TVCACHE can ignore state-preserving tools during longest-prefix matching. In the example, both rollouts share the same stateful prefix $(t_1, t_2)$ (grey blocks) and then invoke only stateless tools (blue blocks) afterward. Although the stateless suffix appears in different orders (e.g., $t_3$ then $t_4$ versus $t_4$ then $t_3$), these calls do not change the sandbox state. Therefore, TVCACHE can perform LPM over only that filtered subsequence without impacting correctness. This increases reuse opportunities in workloads where developers can annotate tool calls effectively as state-preserving or state-modifying.

**Addition to TCG.** On a cache miss under stateful prefix matching, TVCACHE first computes the longest-prefix match using only the state-modifying calls in the rollout history. Let $v$ be the TCG node reached by this matched filtered prefix. The client forks the sandbox snapshot stored at $v$ (or starts from a clean sandbox if no snapshot exists) and replays the remaining tool calls from that point, executing all remaining tools in the suffix. Each tool in the suffix is attached to the last node in the prefix with a state-modifying tool. For example, in Figure 10, $t_3$ and $t_4$ from Rollout 1 attach to $t_2$ even though $t_4$ executes after $t_3$ in Rollout 1. Similarly, $t_3$ and $t_4$ from Rollout 2 attach to $t_2$ even though $t_3$ executes after $t_4$ in Rollout 2.

By indexing TCG nodes using only state-modifying prefixes while replaying state-preserving tool calls directly from cached outputs, TVCACHE increases both cache-hit and LPM rates without sacrificing correctness.

## C. End-to-end evaluation configuration

| Workload | Agent | Batch size | Loss function | Framework |
|---|---|---|---|---|
| terminal-bench (easy) | Qwen3-4B-Instruct-2507 | 4 | GRPO | veRL |
| terminal-bench (med) | Qwen3-4B-Instruct-2507 | 4 | GRPO | veRL |
| terminal-bench (easy) | Qwen3-14B-Instruct | 16 | GRPO | veRL |
| terminal-bench (med) | Qwen3-14B-Instruct | 16 | GRPO | veRL |
| SkyRL-SQL | Qwen2.5-Coder-7B-Instruct | 64 | GRPO | SkyRL |
| EgoSchema | Qwen3-30B-A3B-Instruct-2507 | 4 | Importance sampling | Tinker API (cloud) |

*Table 5.* End-to-end evaluation configuration details for different workloads.

Table 5 summarizes the experiment parameters we used in our end-to-end evaluation.

**Reward.** We use the same reward scheme for all three datasets. A rollout receives a reward of $-1$ if any tool call has an incorrect format, $0$ if all tool call formats are correct but the final answer is wrong, and $1$ if both the tool call formats and the final answer are correct. For terminal-bench, we determine success by running the dataset-provided test scripts. For the SkyRL-SQL benchmark, we compare the rollout's SQL query result to the expected result provided in the dataset. For EgoSchema, we compare the rollout's final multiple-choice answer to the ground-truth answer option.

## D. Tools for EgoSchema

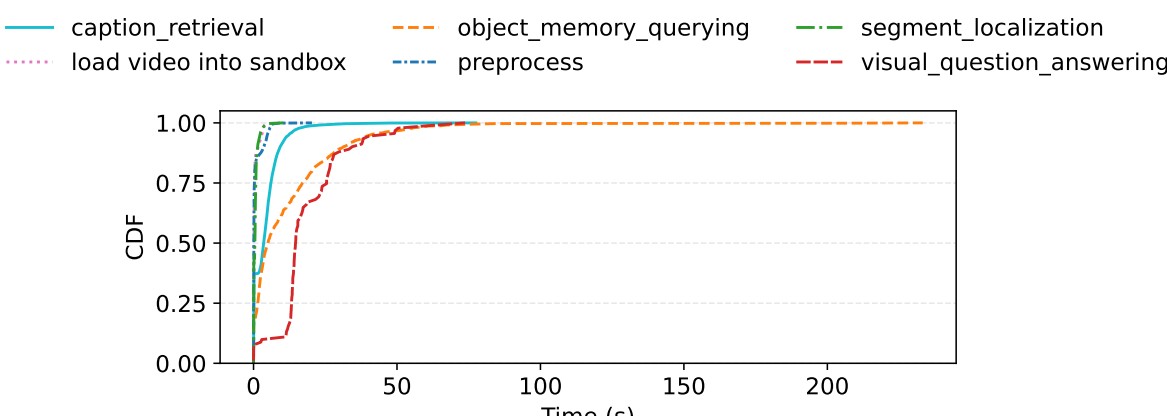

*Figure 11.* Distribution of execution times of all tools in EgoSchema's finetuning process over 5 epochs. Calls to `preprocess` and `load_video` take negligible time because they copy precomputed artifacts or video files into the sandbox, both of which are fast file system operations.

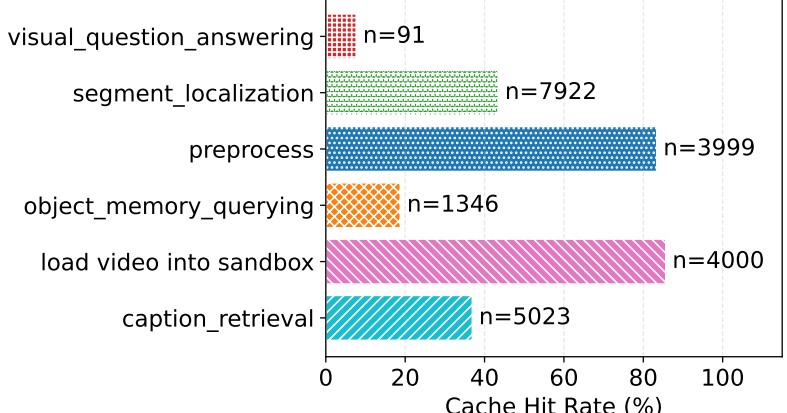

*Figure 12.* Cache hit rates across all tools in the EgoSchema finetuning process across 5 epochs on Qwen-30B. Calls to `preprocess` and `load_video` have the highest hit rate. Calls to `object_memory_querying` and `visual_question_answering` have the lowest hit rate because they use strings as arguments. Here $n$ is the total number of tool call invocations across 5 epochs.

We evaluate TVCACHE on the EgoSchema (Mangalam et al., 2023) dataset, which consists of 3-minute videos with associated multiple-choice questions. Although EgoSchema contains 500 videos with publicly available answers, we use a subset of 100 due to compute constraints. We follow the experimental setup in Table 1 and use Tinker's default loss function.

**Experiment setup.** We host the models required by the VideoAgent tools on a L40S GPU server with 128 CPU cores and 128 GB RAM. Note that Tinker executes the control loop locally on the L40S GPU server but the actual training loop to update weights and inference loop to generate rollouts on the cloud (Lab, 2025).

**Tool descriptions.** A full description of these tools is available in Appendix G. The `preprocess` tool is expensive because it extracts frames from the video, tracks objects, and annotates segments with short descriptions using models on the L40S GPUs. Instead of preprocessing the same video repeatedly, we preprocess all videos in the dataset once before training and reuse the results whenever an agent invokes `preprocess`. We use all the tools as-is except `caption_retrieval`,

which we adapt to use the OpenAI API to generate captions instead of a local model. The modified `caption_retrieval` tool sends the associated frames to the API with a prompt directing the LLM to generate a caption describing the scene in the frames. We extend `caption_retrieval` to increase the diversity of the tools we evaluate.

**Distribution of tool execution times.** Figure 11 shows the distribution of execution times for the tools used in the EgoSchema workload. The `object_memory_querying` tool has the highest execution time because it internally runs an agent loop using an OpenAI model (Fan et al., 2025). However, the agent being trained invokes this tool less frequently than most other tools (Figure 12). The `preprocess`, `load_video`, and `segment_localization` tools are inexpensive because they primarily copy dataset artifacts into the sandbox or perform lightweight lookup operations. The `visual_question_answering` tool has the second-highest execution time because it runs a video question-answering model on the GPU, but gets invoked least frequently because the prompt warns that this tool may hallucinate. The `caption_retrieval` tool is faster than `visual_question_answering` and `object_memory_querying`, and the agent invokes it more frequently than both.

**Cache hit rates.** The cache hit rate is highest for `load_video` and `preprocess` because the prompt specifies that the LLM must execute these two tools before calling any other tool. The LLM learns to do this very early in the first few rollouts, and every subsequent rollout sees a cache hit for both tools. The hit rate for `object_memory_querying` and `visual_question_answering` is low because the arguments to these tools are strings. Even minor differences in text create different nodes in the Tool Call Graph (TCG), even when the arguments have the same meaning (§3). The hit rate of `caption_retrieval` is higher because the arguments are integers and many rollouts of a task generate `caption_retrieval` tool calls with the same segments as arguments. Every cache hit in `caption_retrieval` saves API token costs resulting in $1.67\times$ reduction in total token usage and up to $3\times$ reduction for certain tasks in the dataset.

**Traditional cache for API savings?** While we can cache `caption_retrieval` requests and responses from the OpenAI API in a traditional cache to reduce token usage, retrieving values from this cache leads to incorrect output across rollouts. Rollouts from different tasks can issue identical tool call signatures (*e.g.,* `caption_retrieval(segment_1,` `segment_2)`), yet the underlying video inputs differ across sandboxes, leading to different expected outputs for each task. A conventional caching system that uses the tool call signature as key will incorrectly assume that identical tool calls from rollouts corresponding to different videos will produce the same output. Correct caching would require tagging each tool call with the sandbox state, including the associated video file, which effectively converges to TVCACHE's design.

**Skipping stateless tools.** Of all the tools in this benchmark, only `load_video` and `preprocess` modify the state of the sandbox. We annotate these two tools by setting `will_mutate_state()` to return true, while all other tools (`object_memory_querying`, `segment_localization`, `caption_retrieval`, `visual_question_answering`) are marked as state-preserving with `will_mutate_state()` returning false. TV-CACHE's stateful prefix matching exploits this distinction to significantly improve cache reuse by performing longest-prefix matching over only the subsequence of state-modifying tools in the TCG, as described in §B.

**Example 1: Improved LPM reuse.** Consider two rollouts with tool sequences `load_video`, `preprocess`, `caption_retrieval(0, 10)` and `load_video`, `preprocess`, `segment_localization(...)`. Without stateful prefix matching, these sequences would not match beyond the first two tools, since the third tool differs. However, because `caption_retrieval` and `segment_localization` are both state-preserving, stateful prefix matching filters them out during LPM, recognizing that both rollouts reach the same sandbox state after `preprocess`. The second rollout can thus reuse the cached sandbox from the first rollout's prefix node, executing only `segment_localization` locally.

**Example 2: Improved cache hits with reordering.** Consider two rollouts with different orderings of state-preserving tools: Rollout 1 executes

$$\langle \texttt{load\_video}, \texttt{preprocess}, \texttt{caption\_retrieval}(0, 10), \texttt{visual\_question\_answering}(\ldots, 5) \rangle$$

and Rollout 2 executes

$$\langle \texttt{load\_video}, \texttt{preprocess}, \texttt{visual\_question\_answering}(\ldots, 5), \texttt{caption\_retrieval}(0, 10) \rangle.$$

Without stateful prefix matching, the TCG would store these as distinct paths. Rollout 2's call to `visual_question_answering` would be a cache miss because it appears at a different position in the prefix (after `preprocess`, not after `caption_retrieval`). However, with stateful prefix matching, both rollouts share the same state-modifying prefix `load_video`, `preprocess`, and the TCG indexes both `caption_retrieval` and

visual_question_answering as children of the preprocess node (Figure 10). Rollout 2's calls to both tools are cache hits, as TVCACHE retrieves their cached results regardless of execution order. This optimization increases cache hits because state-preserving tools no longer force rollout histories to diverge in the TCG. TVCACHE indexes TCG nodes only by state-modifying tool calls, while still caching the outputs of state-preserving tools for reuse.

## E. Docker Sandbox design

We build the sandbox for the terminal-bench training loop on top of the open-source harness provided by terminal-bench for agentic benchmarking (The Terminal-Bench Team, 2025). Every task is represented as a set of Docker services in a Docker Compose file which the harness starts and stops before and after the agent executes. The harness internally manages Docker containers using Docker Compose functionality. We implement TVCACHE's sandbox APIs (ToolExecutionEnvironment) using Docker's API within the harness and expose this to the training loop.

**Sandbox server.** We modify the harness to expose APIs to start, stop and fork a sandbox. We reuse the start and stop methods in the harness as-is but implement the forking feature. We expose the sandbox server as an HTTP server to decouple the hardware where the sandboxes execute and the training loop executes. The sandbox containers live on the sandbox server and TVCACHE in the training framework communicates with this server using sandbox IDs to create, delete and fork sandboxes.

**Forking sandboxes.** To implement the forking mechanism, we snapshot a running container and start a new container from that snapshot. We snapshot the container state using Docker's commit API along with the --no-pause flag to avoid blocking the running workload. Moreover, we also store any environment variables modified during execution along with the container's current working directory to match the state of the running container. Finally, we start a new container from the committed image and restore the environment variables and working directory.

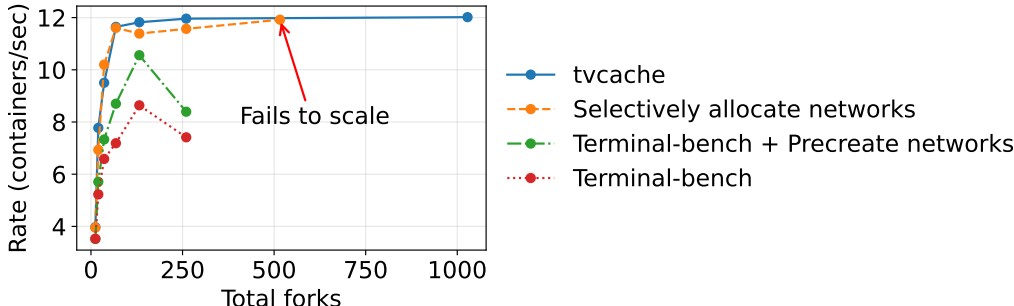

*Figure 13.* Container creation rate as a function of total forks. *terminal-bench* shows the default harness behavior. *terminal-bench + Precreate networks* shows performance when bridge networks are pre-created and reused. *Selectively allocate networks* shows performance when networks are allocated only for compose files that require them. *tvcache* shows performance with selective network allocation and rate-limited container creation.

**Scaling sandboxes.** Training with proactive forking can require creating hundreds of Docker containers concurrently. For example, with a batch size of 16, 4 rollouts per prompt, and 10 cached sandboxes in the TCG, the system may need to fork up to 640 containers in a short time window. However, the default terminal-bench harness does not scale to this level of concurrency. As shown in Figure 13, the baseline terminal-bench configuration achieves limited container creation throughput and degrades rapidly as the number of concurrent forks increases.

**Network creation overhead.** A major bottleneck arises from Docker Compose's default behavior of creating a dedicated bridge network for every sandbox. This overhead significantly limits scalability, as reflected by the *terminal-bench* curve in Figure 13. To address this, we pre-create a pool of bridge networks and reuse them across sandboxes, which improves container creation throughput (*terminal-bench + Precreate networks*).

**Selective network allocation.** Further inspection revealed that many tasks do not require networking at all: only compose files that expose ports or include multiple services depend on an isolated network. A simple check that parses a docker-compose file to count the number of services and exposed ports classifies a task as requiring network or not. By classifying tasks and allocating networks only when required, we further increase throughput and reduce overhead, as shown by *Selectively allocate networks* in Figure 13.

**Rate-controlled forking.** As the number of concurrent fork requests increases, throughput eventually plateaus and container creation becomes unstable. At this point, Docker issues a large number of concurrent requests to create cgroups and other kernel-level resources, causing timeouts when the kernel cannot service system calls quickly enough. Rather than modifying kernel behavior, we exploit the observation that container creation throughput saturates before failures occur. By capping concurrency at this saturation point and enforcing it via a rate-limited forking pipeline, TVCACHE avoids kernel-level contention while sustaining peak throughput, as shown by *tvcache* in Figure 13.

## F. Terminal-bench rollout times

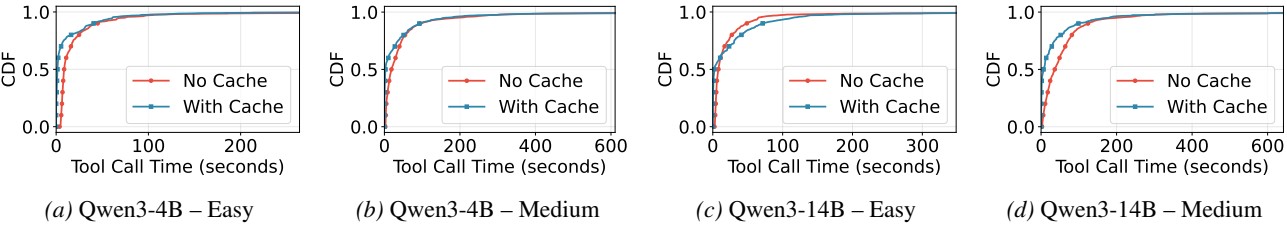

*(a)* Qwen3-4B – Easy     *(b)* Qwen3-4B – Medium     *(c)* Qwen3-14B – Easy     *(d)* Qwen3-14B – Medium

*Figure 14.* The distribution of tool call times for Qwen3-4B and Qwen3-14B on terminal-bench easy and medium tasks, with and without TVCACHE.

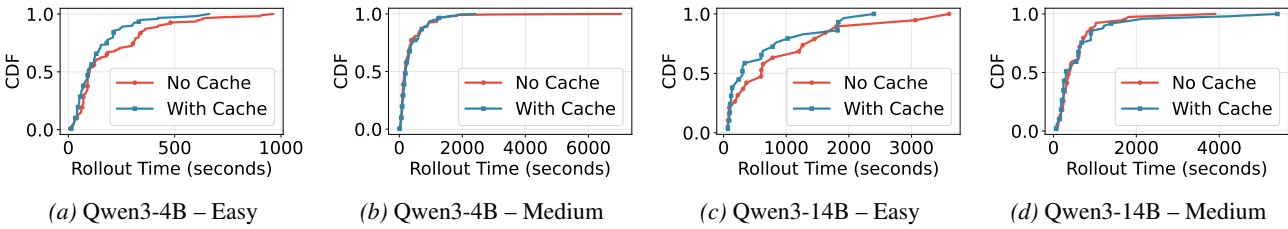

*(a)* Qwen3-4B – Easy     *(b)* Qwen3-4B – Medium     *(c)* Qwen3-14B – Easy     *(d)* Qwen3-14B – Medium

*Figure 15.* Longest rollout time per step for Qwen3-4B and Qwen3-14B on terminal-bench easy and medium tasks, with and without TVCACHE.

TVCACHE reduces overall tool-call time as well as the longest-rollout time per training step. In this section we present additional results for TVCACHE with terminal-bench. Note that all the models shown in this section are from Table 1 and are either Qwen3-4B-Instruct or Qwen3-14B-Instruct models. We omit the instruct label for brevity during discussion.

**Tool call times.** In Figure 14, for all the rollouts throughout the training, we collect the total tool call times across multiple turns in a single step and plot the distribution of these tool call times. For better visualization, we remove the last 1% of tool-call times (tail outliers). As shown in the figure, most tool calls complete quickly, but the distribution has a long tail. However, with TVCACHE, the tool-call time distribution shifts left which indicates the reduction in tool call times. With additional experiments, we found that proactive forking accounts for most of the gains since it removes the container startup and stop overhead for every rollout.

**Longest rollout time per step.** In Figure 15, we plot the time taken by the longest rollout in each step during training. TVCACHE reduces the longest rollout time per step. Moreover, we see higher gains in the easy tasks compared to the medium tasks because the tool calls executed by rollouts to solve medium tasks are longer due to their complexity.

## G. Prompts Used to Train Each Model

In this section, we present the prompts we used to generate rollouts to fine-tune models on terminal-bench, SkyRL-SQL and EgoSchema.

**EgoSchema Prompt.** In the EgoSchema benchmark training prompt, we expose all the tools VideoAgent (Fan et al., 2025) has and add two additional tools to load the video in the sandbox to support concurrent rollouts on different videos.

You are an AI assistant tasked with answering a multiple-choice question related to a video. You will be given a task instruction and the output from previously executed tools. Your goal is to answer the questions by providing batches of tool calls. The question has 5 choices, labeled as 0, 1, 2, 3, 4. The video is sliced into 2-second segments, each with a segment ID starting from zero and incrementing in chronological order. Each segment has a caption depicting the event. There is an object memory that saves the objects and their appearing segments. The object memory is maintained by another agent.

**You have access to the following tools:**

1. `load_video_into_sandbox(video_name)`: Load the video video_name into a sandbox for processing. This must be called before any other video processing operations.

2. `preprocess()`: Preprocess the video in the sandbox by building temporal memory (captions and segment embeddings) and object memory (tracking and re-identification). This must be called after loading a video and before querying or analyzing it.

3. `object_memory_querying(question)`: Given a question about open-vocabulary objects such as "how many people are there in the video?" or "In which segments did the brown dog appear?", this tool will give the answer based on the object memory. The tool uses sub-tools including database querying and open-vocabulary object retrieval.

4. `segment_localization(description)`: Given a textual description, this tool will return the top-5 candidate segments that are most relevant to the description. Use this to find when specific events or actions occur in the video.

5. `caption_retrieval(start_segment_ID, end_segment_ID)`: Given an input tuple (start_segment _ID, end_segment_ID), this tool will retrieve all the captions between the two segments, 15 captions at most. end_segment_ID < start_segment_ID + 15. Use this to get detailed descriptions of what happens in a time range.

6. `visual_question_answering(question, segment_ID)`: Given an input tuple (question, segment_ID), this tool will focus on the video segments starting from segment_ID-1 to segment_ID+1. It will return the description of the video segment and the answer to the question based on the segment.

**CRITICAL:** You cannot proceed with other tools (object_memory_querying, segment_localization, caption_retrieval, visual_question_answering) WITHOUT loading the video first and then preprocessing the video. What this means is that you need to always first load the video and next preprocess it. Only after this completes and you observe it is complete, you can proceed calling other tools.

**ATTENTION:**

1. The segment captions with prefix '#C' refer to the camera wearer, while those with prefix '#O' refer to someone other than the camera wearer.

2. You can use both 'visual_question_answering' and 'object_memory_querying' to answer questions related to objects or people.

3. The 'visual_question_answering' may have hallucination. You should pay more attention to the description rather than the answer in 'visual_question_answering'.

4. Use single quotes on the string arguments of the tools. The input to the tools should not contain any double quotes. If the tool has two arguments, output the arguments in brackets such as ('what is the man doing', 1).

5. It's easier to answer the multiple-choice question by validating the choices.

6. If the information is too vague to provide an accurate answer, make your best guess.

7. Use segment_localization to quickly find relevant parts of the video before using caption_retrieval or visual_question_answering for details.

8. For questions about specific objects or people across the entire video, use object_memory_querying first.

**Question:** {QUESTION}

Your response must be a JSON object that matches this schema. Do not include markdown formatting in your response.

$< JSON\_SCHEMA >$

**SkyRL-SQL Prompt.** In the SkyRL-SQL benchmark training prompt, we provide the model with a database schema, database engine, external knowledge required for the task, instructions, format and an example response.

**Task Overview:**
You are a data science expert. Below, you are provided with a database schema and a natural language question. Your task is to understand the schema and generate a valid SQL query to answer the question within limited turns. You should breakdown the problem, draft your reasoning process, and generate the solution.

**Database Engine:** SQLite

**Database Schema:** {db_details}
This schema describes the database's structure, including tables, columns, primary keys, foreign keys, and any relevant relationships or constraints.

**External Knowledge:** {external_knowledge}

**Question:** {question}

**Instructions:**

1. Make sure you only output the information that is asked in the question. If the question asks for a specific column, make sure to only include that column in the SELECT clause, nothing more.

2. The generated query should return all of the information asked in the question without any missing or extra information.

3. Before generating the final SQL query, please think through the steps of how to write the query. It should include detailed considerations such as analyzing questions, summarizing relevant findings, brainstorming new ideas, verifying the accuracy of the current steps, refining any errors, thinking of how to call SQL tools, and revisiting previous steps.

**Format:**

1. Conduct thinking inside `<think>...</think>` blocks every time you get new observation or information.

2. You can use SQL tool written within a single `<sql>your sql</sql>` block to explore or verify. SQL tool output will be shown as dataframe inside `<observation>...</observation>`. Based on this observation, you can think again and refine.

3. The returned dataframe will be truncated in 50 rows if observation is too long.

4. If you find no further exploration is needed or reaches max turns, you MUST directly provide the final SQL query solution inside `<solution>...</solution>`.

———————————— START OF EXAMPLE ————————————
Question: how many pigs are in the farm?
Database Schema:
Table: animals
- id (INTEGER, PRIMARY KEY)
- species (TEXT)
- age (INTEGER)
- name (TEXT)

`<think>`I am querying how many pigs are in the farm. I will begin by checking if the 'animals' table exists and contains entries with species = 'pig'.`</think>`
`<sql>`SELECT COUNT(*) FROM animals WHERE species = 'pig';`</sql>`
`<observation>`

```
+----------+
| COUNT(*) |
+----------+
|    12    |
+----------+
```

`</observation>`
`<think>`The result indicates that there are 12 pigs in the farm. Since the question asks for how many pigs, I can now output the final SQL as the solution.`</think>`
`<solution>`SELECT COUNT(*) FROM animals WHERE species = 'pig';`</solution>`
———————————— END OF EXAMPLE ————————————

**Terminal-bench Prompt.** In the terminal-bench training prompt, we provide the model with instructions for executing the command on the terminal inside a tmux session, response json schema and the current terminal state along with task specific

instructions and requirements.

You are an AI assistant tasked with solving command-line tasks in a Linux environment. You will be given a task instruction and the output from previously executed commands. Your goal is to solve the task by providing batches of shell commands.

**For each response:**

1. Analyze the current state based on any terminal output provided

2. Determine the next set of commands needed to make progress

3. Decide if you need to see the output of these commands before proceeding

**Instruction:** {TASK_INSTRUCTION}
**Requirements:** {TASK_REQUIREMENTS}

**Note:** You operate directly on the terminal from inside a tmux session. Use tmux keystrokes like `C-x` or `Escape` to interactively navigate the terminal.

One thing to be very careful about is handling interactive sessions like less, vim, or git diff. In these cases, you should not wait for the output of the command. Instead, you should send the keystrokes to the terminal as if you were typing them. You have access to do anything on this machine, assume you can read, write update and delete files, I am giving you complete access to do this. Respond only with json in the given schema, this is the most important thing to do, respond only with the json in the given schema.

**Your response must be a JSON object that matches this schema:**

$< JSON\_SCHEMA >$

Don't include markdown formatting.

**The current terminal state is:**
`root@lambda1:/app#`

