# OpenReview forum: "Tvcache: A Tool-Value Cache for Post-Training LLM Agents"
_ICML.cc/2026/Conference — ICML 2026 regular_

### Official Review · Reviewer_jVkv · 2026-02-19

**Soundness:** 3
**Presentation:** 3
**Significance:** 3
**Originality:** 3
**Overall Recommendation:** 4
**Confidence:** 2

**Summary:**

The authors propose TVCACHE, a stateful tool-value caching mechanism designed to accelerate RL post-training of LLM agents, where slow external tool calls often leave GPUs idle and increase training costs. By organizing observed tool-call histories into a tree and performing longest-prefix matching, the method reuses cached results only when the full interaction history matches, ensuring identical environment states and correctness. Experiments across terminal-based tasks, SQL generation, and video understanding demonstrate up to 70% cache hit rates and a 6.9× reduction in median tool execution time, while maintaining stable reward accumulation during post-training.

**Compliance With Llm Reviewing Policy:**

Affirmed.

**Final Justification:**

Overall, this paper presents an interesting work. The authors have addressed my concerns during the rebuttal, and I am inclined to accept it.

**Key Questions For Authors:**

see Weaknesses

**Limitations:**

yes

**Strengths And Weaknesses:**

The authors design a caching mechanism that enables correct reuse of computation across stateful tool calls during RL post-training. The core idea focusing on the sequence or trajectory of tool calls that produce the current sandbox state, rather than treating individual tool calls in isolation is interesting and well-motivated. However, there are several major concerns:
1.Lack of comparisons with alternative methods. The paper does not provide sufficient baselines, making it difficult to assess the real benefit of modeling tool-call trajectories instead of isolated calls. The authors should include comparisons with other caching or performance optimization strategies to better demonstrate the advantages of their design.
2.Generality across models, hardware, and RL algorithms. It is unclear whether the proposed method is independent of specific models, hardware setups, or RL training frameworks. If it is intended to be general, the authors should evaluate whether different RL training methods affect its performance or cache effectiveness.

Overall, the proposed idea is interesting, but the experimental evaluation is not yet sufficiently comprehensive.

---

> ### Author Rebuttal · Authors · 2026-03-31
>
> Thank you for the constructive review and for the positive assessment of our work. We address your concerns with new experiments and discussion below.
>
> ---
>
> ## W1: Lack of Comparison with Alternative Methods
>
> The most natural baseline for TVCache is a **stateless cache**, keyed on tool name and arguments alone, ignoring sandbox history. We have now run this comparison on terminal-bench across three dimensions:
>
> **Correctness.** A stateless cache returns wrong results on a large fraction of hits:
>
> | Model | Incorrect Hits (Stateless Cache) |
> |---|---|
> | Qwen3-4B-Instruct | **27.4%** |
> | Qwen3-14B-Instruct | **26.1%** |
>
> These bad hits substitute stale tool outputs (e.g., returning pre-patch file contents after the agent has already modified the file), silently corrupting the reward signal.
>
> **RL training performance.** Because over 25% of stateless cache hits are wrong, the agent receives a systematically distorted view of the environment, leading to degraded policy updates. In contrast, TVCache's reward curves exactly match the no-cache baseline (Figure 6). A stateless cache does not offer this guarantee.
>
> **Hit rate.** A stateless cache achieves a higher *raw* hit rate by ignoring state but a large fraction of those hits are incorrect. TVCache's hit rate is lower and fully correct.
>
> Beyond the stateless baseline, other natural performance-optimization strategies are asynchronous tool execution (verl) and simulated tool responses (Kimi K2). Both are complementary to TVCache rather than alternatives, as we discuss in Section 5. TVCache can be combined with either approach to further improve post-training iteration times.
>
> ---
>
> ## W2: Generality Across Models, Hardware, and RL Algorithms
>
> TVCache does not assume or leverage any model- or hardware-specific features.
> We have evaluated TVCache across a broad set of configurations, which we summarize here:
>
> | Dimension | Configurations Evaluated |
> |---|---|
> | Model size | Qwen3-4B, Qwen3-14B, Qwen2.5-7B, Qwen3-30B |
> | Hardware | Self-hosted (2×A100), cloud (8×A100), Tinker API |
> | Sandbox type | Docker containers, SQLite, filesystem |
> | RL framework | verl, SkyRL, Tinker (RL-as-a-service) |
> | Task domain | Terminal (bash), SQL, video understanding |
>
> These configurations show that TVCache is model- and hardware-agnostic by design. It integrates via the `ToolCallExecutor` interface, which any RL training loop can implement (Section 3.4).
>
> While the performance gains from caching improve when the RL algorithm involves multiple rollouts in a batch (as in GRPO and its variants), TVCache also leverages tool call
> redundancy *across training iterations* and therefore demonstrates performance gains even with a single rollout per iteration. To show this, we ran new terminal-bench and
> SkyRL-SQL experiments varying the number of parallel rollouts per task:
>
> | Parallel Rollouts / Task | terminal-bench Avg. Cache Hit Rate | SkyRL-SQL Avg. Cache Hit Rate
> |---|---|---|
> | 1 | 10.1% |23.4%
> | 2 | 13.1% |23.3%
> | 4 | 17% |27.8%
> | 8 *(GRPO, as in paper)* | 20.2% |33.4%
>
> Even at 1 rollout per task, TVCache achieves a **10.1% and 23.4% hit rate** via cross-epoch reuse of tool calls, yielding up to 35% reduction in end-to-end training time. This shows that TVCache is not specific to GRPO, and while its utility improves with parallel rollouts, it can be beneficial even when there are no parallel rollouts.
>
> ---
>
> We will incorporate the stateless comparison and the parallelism sweep into the revised paper. We hope these results address both concerns and are happy to discuss further.

---

> > ### Author Rebuttal · Reviewer_jVkv · 2026-04-01
> >
> > Thank you for response. My concerns have been partially addressed, and I will maintain my current score.

---

### Official Review · Reviewer_JLzu · 2026-03-03

**Soundness:** 4
**Presentation:** 3
**Significance:** 3
**Originality:** 3
**Overall Recommendation:** 5
**Confidence:** 2

**Summary:**

This work first identifies tool calling as one bottleneck in RL post-training. Then, it proposes a stateful tool call caching mechanism that: 1, targets hitting both current tool call and previous traces; 2, selectively stores sandbox snapshots. Through experiments on three benchmarks, it shows that though the hitting mechanism is more stringent, it still achieves decent hit rate (Figure 5); it can significantly reduce tool call latency (as shown in Table 2); 3, RL performance is not affected by this caching mechanism (Figure 6).

**Compliance With Llm Reviewing Policy:**

Affirmed.

**Final Justification:**

I thank the authors for their detailed responses. My concerns are now welled addressed, and I believe the proposed method is a good contribution to the community. I decided to raise my score.

**Key Questions For Authors:**

I will summarize my questions mentioned in above:
1, What are the overall training time with and without the proposed stateful caching mechanism;
2, If using stateless caching, how would the overall latency, hit rate, and RL training performance change?

**Limitations:**

- Memory footprint consumption: while it is analyzed in section 4.6 that memory overhead is modest, on tasks where the sandbox environments could be complicated, the proposed method might induce significant memory footprint.

**Strengths And Weaknesses:**

# Strength
1, Overall, this work studies a very interesting and important problem. The proposed method could facilitate more efficient agentic RL training, which is an important contribution to both academia and the industry.

2, The proposed stateful caching mechanism avoids errors from caching mechanisms induced by stateless mechanisms; it is also supported by empirical results in Figure 6.

3, Proposed method also obtains good empirical results in hit rates, and s/call.

# Weakness
1, My first concern is with the effect on overall training time. While Table 2 shows that per tool call latency is significantly reduced, it does not reflect how much the proposed caching mechanism could save the overall training latency, especially considering that the proposed graph based search and sandbox snapshot mechanisms would induce extra latency.

2, While the stateful caching mechanism makes sense to me, its not clear to me how the stateful mechanism is compared to stateless mechanism. It would be important to include a stateless baseline to answer the following questions:
- How does stateless caching affect the RL performance?
- What is the hit rate of stateless baseline?
- How much latency can stateless caching reduce?
Considering that being stateful is a key contribution of the proposed method, including a stateless baseline and answering the above questions would make the submission much stronger.

---

> ### Author Rebuttal · Authors · 2026-03-31
>
> Thank you for the careful and positive review. We address the two concerns raised in your review with new experimental results.
>
> ---
>
> ## W1: Effect on Overall Training Time
> We agree that per-tool-call speedup (Table 2 in the paper) does not by itself quantify end-to-end training time savings. We have run new experiments to measure end-to-end improvement in post-training time using TVCache. On terminal-bench tasks of medium hardness, TVCache reduces end-to-end post-training time for the Qwen3-14B model by 2.2 hours — a 35% reduction. Because this configuration generates tool call timeouts whose durations can vary across runs, we report the median per-step longest rollout time as our comparison metric, which provides a robust measure of end-to-end savings between the cached and uncached configurations.
>
> Two factors govern the translation from per-call speedup to end-to-end savings. First, since batch completion time is bottlenecked by the *slowest* rollout, reductions in long-tail tool calls (P99 >92% of rollout time on terminal-bench) have an outsized effect. Second, TVCache's own overhead is small: cache lookup latency is under 10ms at P95 (Figure 8a), and proactive forking offloads sandbox creation to background threads. The net result is that end-to-end training time decreases meaningfully even though not every rollout benefits equally. Figure 7 in the paper shows per-rollout and per-batch time reductions for EgoSchema, illustrating this dynamic.
>
> We will add the end-to-end training time comparison table for all evaluation scenarios in the revised paper.
>
> ---
>
> ## W2: Stateless Caching Baseline
> The reviewer has highlighted that we have not explicitly shown what is wrong with using a stateless cache. This is an important comparison and one we have now run. We evaluated a stateless cache on terminal-bench, keyed only on tool name and arguments, ignoring sandbox history, across three dimensions:
>
> **Correctness.** A stateless cache returns wrong results on a large fraction of hits:
>
> | Model | Incorrect Hits (Stateless Cache) |
> |---|---|
> | Qwen3-4B-Instruct | **27.4%** |
> | Qwen3-14B-Instruct | **26.1%** |
>
> These bad hits silently substitute stale tool outputs (e.g., returning the pre-patch content of a file after the agent has already modified it), corrupting the reward signal and gradient updates. This confirms that stateless caching is not a safe approximation — it is incorrect for stateful environments.
>
> **RL training performance.** As a consequence of the above corruption, we observe that a stateless cache degrades post-training reward. Because over 25% of cache hits return wrong results, the agent receives a systematically distorted view of the environment, leading to miscalibrated rewards and degraded policy updates. TVCache, by contrast, produces reward curves that exactly match the no-cache baseline (Figure 6).
>
> **Hit rate.** A stateless cache does achieve a higher *raw* hit rate than TVCache, since it ignores sandbox state and matches on tool signature alone. However, as shown above, a large fraction of those hits are incorrect. TVCache's hit rate (shown in Figure 5 of the paper) is lower but every hit is guaranteed to be correct.
>
> **On memory overhead.** The reviewer rightly notes that complex sandbox environments could increase memory footprint. TVCache bounds this via the sandbox budget mechanism (Section 3.3 of the paper): each task specifies a maximum number of active sandboxes, and TVCache evicts subtrees with low expected reuse when the budget is exceeded. For tasks with large or expensive sandboxes, practitioners can reduce the budget to control memory at the cost of some cache hit rate. We will add a discussion of this trade-off to the revised paper.
>
> ---
>
> We hope these results, particularly the stateless corruption rates and end-to-end training time measurement, address both your concerns. We are happy to discuss further and will incorporate all new results into the revised paper.

---

> > ### Author Rebuttal · Reviewer_JLzu · 2026-04-03
> >
> > I thank the authors for their detailed responses. My concerns are now welled addressed, and I believe the proposed method is a good contribution to the community. I decided to raise my score.

---

### Official Review · Reviewer_gwmy · 2026-03-07

**Soundness:** 3
**Presentation:** 3
**Significance:** 2
**Originality:** 2
**Overall Recommendation:** 4
**Confidence:** 3

**Summary:**

This paper investigates an important efficiency bottleneck in the reinforcement learning (RL) post-training of LLM agents: the time spent waiting for external tool executions while GPU resources remain idle. The authors propose TVCACHE, a stateful tool-value caching system that allows to reuse the results of previous tool calls to speedup new rollouts. Because many environments are stateful (and therefore tools called in different states lead to different outputs) TVCACHE organizes observed tool sequences into a Tool Call Graph (TCG) that helps track the MDP. It uses  longest-prefix matching to identify hits and to return or extend previously cached results. TVCACHE also includes additional optimizations like selective state snapshotting (essentially snapshotting the environment state when it will help reduce overall time despite the overhead of doing so). The authors showcase TVCACHE in different environments and on top of different RL frameworks, demonstrating in all cases a training speedup with no degradation in performance.

**Compliance With Llm Reviewing Policy:**

Affirmed.

**Final Justification:**

I have decided to update my score as the authors have partially resolved my concerns regarding TVCACHE's performance across different RL algorithms and for probabilistic environments. While the contribution remains primarily at the systems level, this is a domain defined to be of interest by the conference. The authors have also provided evidence to better support the need for their solution. I still believe however, that the level of the contribution of impact of this work deserves a weak accept.

**Key Questions For Authors:**

- The authors primarily evaluate TVCACHE using GRPO, which naturally produces groups of rollouts starting from the same prompt, leading to high tool-call redundancy. Wouldn't TVCACHE become less useful for algorithms not based on GRPO, such as standard Actor-Critic algorithms like PPO, where there may be fewer concurrent rollouts per prompt?

**Limitations:**

The authors include a dedicated Impact Statement that adequately addresses both potential societal risks and the broader positive impacts of their work.

**Strengths And Weaknesses:**

Strengths:
- The paper is well-written and easy to follow, clearly articulating both the motivation behind the system and the technical hurdles of stateful caching.
- It tackles a high-impact infrastructure problem, providing a way to reduce idle time during the RL post-training phase, therefore accelerating the process.
-  TVCACHE is complementary to most existing RL post training frameworks, and therefore easy to adopt.
- The evaluation is comprehensive, spanning different types of sandboxes (Docker, SQLite, and file systems) and infrastructure types (self-hosted vs. cloud API)

Weaknesses:
- The primary contribution is systems-level optimization. While it enables better ML research, the paper does not propose new RL algorithms or architectures or sheds light over research questions in the field.
- The proposed caching mechanism seems strictly limited to deterministic environments, while it is not uncommon for RL environments to be non-deterministic (for example, a navigation task involving a web browser or any multi-agent system). This limits its applicability.

---

> ### Author Rebuttal · Authors · 2026-03-31
>
> We thank you for the constructive review, and for recognizing the importance and impact of tool-calling as a post-training bottleneck. Our work indeed addresses this inefficiency with a cache that is performant and yet "easy to adopt" in various post-training scenarios. We acknowledge the limitations raised in the review and discuss them in detail below.
>
> ## W1: Systems-level optimization as a contribution
>
> Tool-calling latency is a well-recognized source of inefficiency in post-training language agents. For example, [Kimi K2](https://arxiv.org/pdf/2507.20534) uses a "sophisticated tool simulator" to execute tool calls and bound rollout times, with "real execution sandboxes for scenarios where authenticity is crucial". [GLM-5](https://arxiv.org/pdf/2602.15763) notes that "tail latency is often driven by small-BS stragglers (e.g., rare long contexts, complex multi-turn reasoning, **tool-heavy traces**)" (emphasis ours).
>
> TVCache addresses this inefficiency in a principled manner, with reproducible gains across multiple frameworks (verl, Tinker), models, benchmarks (Sky-RL, Terminal-Bench, EgoSchema) and GPU hardware configurations. TVCache's impact compounds across hundreds of training iterations:
>
> | Metric                                                  | Value                        |
> | ------------------------------------------------------- | ---------------------------- |
> | Reduction in tool call overhead, terminal-bench (mean / P99) | 43% / >92% of rollout time   |
> | Median per-tool-call speedup                            | up to **6.9X**               |
> | End-to-end training time reduction                      | up to **35%** |
> | OpenAI API token cost reduction (EgoSchema)             | up to **3X**                       |
> | Cache hit rate (EgoSchema)                   | **73.9%**                    |
>
> The up to **3X API cost reduction** benefits practitioners using cloud-hosted tools, with *zero degradation in reward accumulation* (Figure 6).
>
> These efficiency gains are likely to grow as tool-calling becomes more complex: for example, the tool calls in [agentic vision](https://openai.com/index/thinking-with-images/) involving zooming and cropping images or video frames. Indeed, our work fits under the [ICML Call For Papers](https://icml.cc/Conferences/2026/CallForPapers) category of "machine learning systems (improved implementation and scalability, hardware, libraries, distributed methods, etc.)".
>
> ---
>
> ## W2: Non-deterministic Environments
>
> Non-determinism in the outcome of a tool call may arise from:
>
>   1. A change in *observable state* (e.g., a database table that is part of the post-training setup is written to, which changes the results of subsequent reads).
>
>   2. A change in *unobservable state* (e.g., Google's search index changes, which changes the result of subsequent calls to a web search tool).
>
>   3. True stochasticity (e.g., the tool calls another language model, whose output is stochastic).
>
> TVCache addresses (1) by keying on the state, since state modifications will produce a fork in the tool call graph. To address (2) and (3), we have now implemented a **time-to-live (TTL) mechanism** that invalidates cached entries after a configurable duration. Practitioners can tune the TTL and recover caching benefits within each TTL validity window while preventing stale cache values. This new functionality is now a part of the TVCache codebase which we have released along with this response: https://github.com/TVCache/TVCache. We will describe details of the TTL mechanism in the revised paper.
>
> ## W3: Fewer Concurrent Rollouts Per Prompt
>
> The tool call graph of TVCache persists *across iterations*, so cross-iteration reuse of tools leads to cache hits regardless of the degree of intra-iteration parallelism. To show this empirically, we ran new terminal-bench and SkyRL-SQL experiments in which we vary the number of parallel rollouts per task:
>
> | Parallel Rollouts / Task | terminal-bench Avg. Cache Hit Rate | SkyRL-SQL Avg. Cache Hit Rate
> |---|---|---|
> | 1 | 10.1% |23.4%
> | 2 | 13.1% |23.3%
> | 4 | 17% |27.8%
> | 8 *(GRPO, as in paper)* | 20.2% |33.4%
>
> Even at 1 rollout per task, TVCache achieves a **10.1% and 23.4% hit rate** via cross-iteration reuse. Across experimental evaluation using different size models (Qwen3-4B, Qwen3-14B) TVCache reduces end-to-end training time by up to **35%**.
>
> We hope our response and new results address your concerns. We are happy to discuss further.

---

> > ### Author Rebuttal · Reviewer_gwmy · 2026-04-01
> >
> > Thank you for your responses. My concerns regarding non-determinism and performance across different RL algorithms and environments are partially resolved, and I will update my score accordingly.

---

### Decision · Program_Chairs · 2026-04-30

**Decision:**

Accept (regular)

**Comment:**

TVCache addresses a critical infrastructure bottleneck in agentic RL post-training: the idle GPU time caused by slow external tool executions. Recognizing that naive stateless caching is fundamentally unsafe in stateful environments, the paper proposes a Tool Call Graph with longest prefix matching to guarantee correct tool value reuse. The empirical results are substantial, demonstrating up to 35% end-to-end training time reductions and 3x API cost savings across diverse workloads, without degrading reward signals.

The initial reviews were highly positive (scores of 5, 4, 4), with unanimous praise for the paper’s strong motivation and easy-to-adopt design. The primary concerns centered on generality beyond GRPO, the absence of a stateless baseline, and a lack of comparisons with alternative optimization strategies.

The rebuttal substantially strengthened the paper’s empirical foundation. Most notably, the authors provided a crucial experiment demonstrating that over 25% of stateless cache hits silently corrupt reward signals. This effectively shifted the narrative, establishing that stateful caching is not merely a performance heuristic, but a strict correctness requirement. Furthermore, the authors provided missing end-to-end metrics, introduced a TTL mechanism for non-deterministic environments, and demonstrated cross-iteration reuse to decouple the method from GRPO. Two reviewers raised their scores accordingly. Reviewer jVkv requested AC mediation regarding remaining generality concerns; after reviewing the discussion, I found the authors’ clarifications regarding the system’s boundary conditions satisfactory.

The main lingering concern is whether a pure systems optimization carries sufficient conceptual novelty for ICML. However, I am advocating for a weak accept because the paper excels precisely where it matters most for this type of work: it identifies a rapidly growing bottleneck, proves that naive solutions are mathematically unsafe, and delivers a highly practical, reproducible system with undeniable empirical gains. The combination of correctness guarantees and significant cost reductions outweighs the reliance on established caching heuristics.